# Design of Power/Ground Noise Suppression Structures Based on a Dispersion Analysis for Packages and Interposers with Low-Loss Substrates

**DOI:** 10.3390/mi13091433

**Published:** 2022-08-30

**Authors:** Youngwoo Kim

**Affiliations:** Nara Institute of Science and Technology, Division of Information Science, Ikoma-shi 630-0192, Japan; youngwoo@is.naist.jp; Tel.: +81+743-72-5953

**Keywords:** electromagnetic bandgap (EBG), interposers, low-loss substrates, noise suppression structures, packages, power delivery network (PDN), power/ground noise

## Abstract

In this study, power/ground noise suppression structures were designed based on a proposed dispersion analysis for packages and interposers with low-loss substrates. Low-loss substrates are suitable for maintaining signal integrity (SI) of high-speed channels operating at high data rates. However, when the power/ground noise is generated in the power delivery network (PDN), low-loss substrates cannot suppress the power/ground noise, thereby causing PDN-induced crosstalk and various power integrity (PI) issues. To solve these issues, noise suppression structures generating electromagnetic bandgap were proposed and designed. The mechanism of the proposed structures was examined based on a proposed dispersion analysis. The proposed structures were designed and fabricated in glass interposer test vehicles, and the effectiveness of the structures on power/ground noise suppression was experimentally validated by measuring the noise suppression band. The proposed dispersion analysis was also verified by comparing the derived noise stopband edges ($f_{L}$ and $f_{U}$) with electromagnetic (EM) simulation and experimental results, and they all showed good agreement. Compared to EM simulation, the proposed method required smaller computational resources but showed good accuracy. Using the proposed dispersion analysis, various power/ground noise suppression bands were designed considering the applications and design rules of packages and interposers. With measurements and EM/circuit simulations, the effectiveness of the designed structure in maintaining SI/PI was verified. By adopting the designed structures, the noise transfer properties in the PDN were suppressed in the target suppression frequency band, which is key for PI design. Finally, it was verified that the proposed structures were capable of suppressing power/ground noise propagation in the PDN by analyzing PDN-induced crosstalk in the high-speed channel.

## 1. Introduction

Transistor scaling based on Moore’s law is facing a limit. At the same time, the realization of electrical systems with wide bandwidth, superior performances, small form factor, low power consumption, and reduced manufacturing cost has been a continuous challenge. System scaling based on through-silicon via and interposer technologies is a promising solution for current industrial challenges [1,2,3,4,5]. Recently, silicon interposers have been widely used to integrate graphic processing unit (GPU) and high bandwidth memory (HBM) to form 2.5-dimensional (2.5-D) systems to realize over terabyte per second (TB/s) system bandwidths for extremely high-performance computing [6,7]. However, reducing the manufacturing cost of silicon interposers is still difficult due to the throughput issues associated with limited wafer dimension. Even though silicon interposer-based integration and packaging provides promising solutions, reducing the manufacturing cost is difficult due to the following reasons: throughput issues associated with limited wafer dimension and additional fabrication steps to isolate metal layers from the conductive silicon substrate. Moreover, the finite conductivity of the silicon substrate can generate signal integrity (SI) issues in the high-frequency range, which may limit high-speed digital signaling, radio frequency (RF) applications, and mixed-signal integrations [8]. Because of these limitations, packages and interposers with low-loss substrates have gained attention as a long-term alternative to silicon interposers.

Packages and interposers with low-loss substrates have been continuously developed. Glass substrates have several advantages, such as dimensional stability, coefficient of thermal expansion (CTE) similar to silicon, smooth surface, submicron metallization, double-sided build-up lamination, and superior electrical resistivity enabling low insertion loss up to millimeter (mm)-wave range [9,10,11]. Recently, glass substrates have been widely adopted for various packaging applications, such as bandpass filters for 5G communication [12], MEMS [13], millimeter-wave radar for autonomous driving [14], and antennas [15,16]. Fan-out packages with low-loss materials have also been widely adopted [17]. Ultrathin, high-permittivity, and low-loss substrates have been released in the market for printed circuit boards (PCBs), which can also be embedded in packages to realize embedded decoupling capacitors [18,19]. These substrate materials are fabricated from panels that are much larger in dimension compared to silicon wafers. Therefore, these substrates also have the potential to reduce the fabrication cost due to increased yield. The advantages are depicted in Figure 1.

Despite these advantages and potentials, low-loss substrates are vulnerable to power/ground noise generated in the power delivery network (PDN) [20]. Low-loss substrates are excellent for high-frequency signaling, but at the same time, it is difficult to suppress power/ground noise induced in the PDN. The noise can be induced in the PDN due to various reasons, such as simultaneous switching noise (SSN), return current discontinuity of the signal via penetration of the PDN, and coupling from other signal/power/ground interconnections. When the power/ground noise is generated, it propagates along the PDN without suppression, causing various SI/PI (power integrity) and electromagnetic interference (EMI) issues. These issues are shown in Figure 2. When designing a PDN, the allowed power/ground noise margin is becoming tighter because the operating voltages of electrical systems are continuously decreasing to realize low power consumption [21]. Moreover, due to recent trends requiring system-in-package, heterogeneous integration, and mixed modes, various noises are generated in the broadband frequency range. To maximize the advantages of low-loss substrates, power/ground noise must be suppressed and effective suppression bands should be analyzed in advance.

Decoupling capacitor arrays and schemes may be insufficient for achieving broadband power/ground noise suppression. Moreover, assembling various decoupling capacitors to achieve broadband suppression can increase the lateral dimensions of packages and interposers, which directly affects the fabrication yield. For security applications, exposed decoupling capacitors attached to the cryptographic core PDN cause electromagnetic (EM) information leakages and security issues [22,23]. Various electromagnetic bandgap (EBG) structures that are mostly embedded inside the PDN have been proposed and validated to achieve wideband power/ground noise suppression [24,25,26,27,28,29,30,31]. Considering recent trends requiring broadband noise suppression without dramatically affecting lateral dimensions of the package/interposer and unexposed areas for some applications, adopting the EBG structure embedded in the PDN is one of the most promising solutions to solve the power/ground noise issues associated with low substrate loss. However, such structures have not been widely developed and applied for packages and interposers with low-loss substrates. Therefore, an efficient design method considering design parameters and material properties is desired. Using the method, noise suppression structures should be developed and verified.

In this study, power/ground noise suppression structures were designed based on a proposed dispersion analysis for packages and interposers with low-loss substrates. The mechanism of noise suppression was thoroughly examined based on the proposed dispersion analysis. The structures were designed and fabricated in glass interposer test vehicles, and the effectiveness of the structures was experimentally validated by measuring the noise suppression band. The proposed dispersion analysis was also verified by comparing derived noise stopband edges ($f_{L}$ and $f_{U}$) with electromagnetic (EM) simulation. It was confirmed that $f_{L}$ and $f_{U}$ estimations based on the proposed analysis method showed good agreement with those acquired from experiments and simulations. Compared to EM simulation, the proposed method required smaller computational resources but showed good accuracy, which is suitable for early design stages. Using the proposed dispersion analysis, various power/ground noise suppression bands were designed considering the applications and design rules of packages and interposers. With measurements and EM simulations, the effectiveness of the designed structure in maintaining SI/PI was verified. Finally, it was shown that the proposed structures were capable of suppressing power/ground noise propagation in the PDN by analyzing PDN-induced crosstalk in the high-speed channel.

## 2. Proposed Dispersion Analysis: Mechanism of Noise Suppression Band Formation and Stopband Edge Estimation

In this section, a dispersion analysis is proposed to explain the mechanism of noise suppression band formulation in the PDN. The proposed dispersion analysis is also capable of deriving noise suppression (stopband) edges. The proposed dispersion analysis is based on a transmission line (TL) theory and mathematics. Compared to full EM simulations, which require heavy computational resources, the proposed method can efficiently estimate the suppression band. In addition, the impacts of material properties and design rules on the suppression band can be easily understood.

Compared to mesh or grid-type PDN structures in silicon interposers, plane-type PDN can be fabricated in packages and interposers with low-loss substrates, such as glass, ceramic, and organic materials [9,18]. In the plane-type PDN, the power/ground noise propagates in forms of transverse electromagnetic (TEM) and quasi-TEM modes. The PDN becomes a transmission line (TL) for the noise wave. To suppress noise propagation, a noise suppression band can be formed by designing certain repetitive structures generating lumped capacitance (*C*) and inductance (*L*). The band must be analyzed and engineered to cover the target noise band. In this study, power/ground noise propagation is analyzed in the +*x-*direction ($N\left(x\right)=e^{-jk_{x}x}a_{x}$). *k_x_* is an effective phase constant defined as *k_x_* = α*_x_*+ *j*β*_x_*, where α*_x_* becomes an attenuation constant and β*_x_* is a propagation constant of the power/ground noise wave propagating in the +*x*-direction. Let the size of the repetitive structure formed in the PDN be *W*_u_. A two-dimensional periodic structure can be reduced into a one-dimensional array of unit cells (size of *W*_u_) by placing perfect magnetic conductor (PMC) walls at *y* = ±*W*_u_/2, as depicted in Figure 3. This assumption and dimension reduction can be applied as the PMC wall can be located anywhere where there is a zero tangential magnetic (*H*) field [31,32,33]. By adopting the PMC boundary condition, the noise suppression structure and the PDN can be modeled as TL with lumped *C* and *L*. In the following sections, detailed equations for *C* and *L* are provided considering the design, physical dimensions, and material properties.

In typical advanced packages and interposers, build-up layers exist in between the substrate and metal layers (ML), as shown in Figure 3. The power/ground noise propagating in the form of the quasi-TEM mode can be modeled into a TL. The TL has a thickness of *t*_PDN_, which is summation of the substrate thickness (*t*_sub_) and the thickness of two build-up layers (*t*_bu_). The dielectric layers in between power and ground planes can be modeled as a single dielectric mixture layer represented with the effective complex dielectric constant (ε_mix_(ω)) shown in the following equation:(1)$$\varepsilon_{\mathrm{mix}}\left(\omega \right)=\frac{\varepsilon_{\mathrm{sub}}\left(\omega \right)\varepsilon_{\mathrm{bu}}\left(\omega \right)}{q\varepsilon_{\mathrm{sub}}\left(\omega \right)+\left(1-q\right)\varepsilon_{\mathrm{bu}}\left(\omega \right)}$$
where *q* is the volume fraction of the dielectric layers; $\varepsilon_{\mathrm{sub}}\left(\omega \right)$ and $\varepsilon_{\mathrm{bu}}\left(\omega \right)$ are complex permittivity of the substrate and build-up layer, respectively [34]; and ω is the angular frequency (2πf). Real parts of the $\varepsilon_{\mathrm{mix}}\left(\omega \right)$, $\varepsilon_{\mathrm{sub}}\left(\omega \right),$ and $\varepsilon_{\mathrm{bu}}\left(\omega \right)$ are defined as $\varepsilon_{r,\mathrm{mix}}$, $\varepsilon_{r,\mathrm{sub}},$ and $\varepsilon_{r,\mathrm{bu}}$, respectively.

Characteristic impedance (*Z*_0, PDN_) and phase constant (*β*_PDN_) of the TL without *C* and *L* can be expressed as follows:(2)$$Z_{0,\mathrm{PDN}}=\frac{\eta_{0}}{\sqrt{\varepsilon_{r,\mathrm{mix}}}}\;\frac{t_{\mathrm{PDN}}}{W_{u}},\;t_{\mathrm{PDN}}=t_{\mathrm{sub}}+2t_{\mathrm{ub}}$$
and
(3)$$\beta_{\mathrm{PDN}}=\sqrt{\varepsilon_{r,\mathrm{mix}}}\omega /c$$
where η_0_ is the wave impedance of free space, and *c* is the speed of light in a vacuum.

To derive the dispersion equations to estimate the noise stopband, *ABCD* parameters of the TL shown in Figure 3 is analyzed. The unit cell of the TL has an effective phase constant *k_x_*. Compared to the silicon, the substrate and build-up layer of the target study have much lower loss factors. Therefore, lossless condition is adopted for an efficient calculation during the dispersion analysis. The ABCD parameters of the TL’s unit cell can be expressed as follows:(4)$$\begin{array}{c} \left[\begin{array}{cc} A_{u} & B_{u}\\ C_{u} & D_{u} \end{array}\right]=\left[\begin{array}{cc} \cos\left(k_{x}W_{u}\right) & jZ_{0,u}\sin\left(k_{x}W_{u}\right)\\ jZ_{0,u}{}^{-1}\sin\left(k_{x}W_{u}\right) & \cos\left(k_{x}W_{u}\right) \end{array}\right]\\ =\left[\begin{array}{cc} \cos\left(\beta_{\mathrm{PDN}}\frac{W_{u}}{2}\right) & jZ_{0,\mathrm{PDN}}\sin\left(\beta_{\mathrm{PDN}}\frac{W_{u}}{2}\right)\\ jZ_{0,\mathrm{PDN}}{}^{-1}\sin\left(\beta_{\mathrm{PDN}}\frac{W_{u}}{2}\right) & \cos\left(\beta_{\mathrm{PDN}}\frac{W_{u}}{2}\right) \end{array}\right]\times \left[\begin{array}{cc} 1 & 0\\ Y & 1 \end{array}\right]\times \left[\begin{array}{cc} \cos\left(\beta_{\mathrm{PDN}}\frac{W_{u}}{2}\right) & jZ_{0,\mathrm{PDN}}\sin\left(\beta_{\mathrm{PDN}}\frac{W_{u}}{2}\right)\\ jZ_{0,\mathrm{PDN}}{}^{-1}\sin\left(\beta_{\mathrm{PDN}}\frac{W_{u}}{2}\right) & \cos\left(\beta_{\mathrm{PDN}}\frac{W_{u}}{2}\right) \end{array}\right] \end{array}$$
where Y *= j*ω*C/(*1 − $\omega^{2}$*LC)*, and $Z_{0,u}$ is the characteristic impedance of TL with the structure inducing *L* and *C.* Among the four parameters, *A*_u_ is the simplest. By analyzing *A*_u_, a dispersion equation is derived as follows:(5)$$A_{u}=\cos\left(k_{x}W_{u}\right)=\cos\left(\beta_{\mathrm{PDN}}W_{u}\right)+j\frac{Z_{0,\mathrm{PDN}}Y}{2}\sin\left(\beta_{\mathrm{PDN}}W_{u}\right)$$

From (5), the effective phase constant *k_x_* is derived as follows:(6)$$k_{x}=\frac{1}{W_{u}}\cos^{-1}\left[\cos\left(\beta_{\mathrm{PDN}}W_{u}\right)-\frac{\omega \;Z_{0,\mathrm{PDN}}\;C}{2\left(1-\omega^{2}LC\right)}\sin\left(\beta_{\mathrm{PDN}}W_{u}\right)\right]$$

In this study, the lossless condition has been assumed. However, the effective phase constant *k_x_* becomes a complex number. The effective phase constant has an imaginary part when the argument of the inverse cosine function in (6) is outside the interval [−1, 1]. The noise stopband is formed at the frequency range where the imaginary part of (6) is nonzero and changes dramatically. The wave becomes evanescent at this frequency range.

To graphically explain the noise propagation characteristics, dispersion diagrams are estimated from (6) and plotted in Figure 4 as an example. The design parameters and values to derive dispersion diagrams shown in Figure 4 are summarized in Table 1. Values are carefully chosen to derive diagrams that can graphically support the method and concept. Without noise suppression structures in the PDN, the propagation constant of the power/ground noise has a constant slope ($2\pi \sqrt{\varepsilon_{r,\mathrm{mix}}}/c)$ in the frequency domain, which is nondispersive. When the structure forming the *L* and *C* is inserted in the PDN, the power/ground noise propagating in the +*x*-direction ($e^{-jk_{x}x}$) will experience sudden attenuation at certain frequency band. Let −*jk_x_* = −α*_x_*−*j*β*_x_*, and α*_x_* becomes an attenuation constant and β*_x_* represents propagation constant of the power/ground noise wave. In this case, β*_x_* is periodic due to periodicity of the inverse cosine function.

As can be seen from Figure 4, there are frequency bands where the slope of β*_x_* becomes closer to zero (or β*_x_* is at the Brillouin zone boundaries) or the attenuation constant α*_x_* becomes nonzero. These frequency bands are theoretical noise bandgap or electromagnetic bandgap (EBG) where power/ground noise is suppressed and cannot propagate. However, except for the fundamental stopband, the attenuation constant for other bands are too small to suppress noise propagation.

In this study, the dispersion equations are calculated and plotted under the assumption of lossless substrate and build-up layer. In reality, there always exists attenuation of the wave associated with dielectric loss. Usually, noise suppression bands are valid if they can achieve −40 dB or lower isolation characteristics. Therefore, theoretical upper bands marked in Figure 4 cannot play a role as a suppression band. A band that can suppress power/ground noise will exist inside the fundamental stopband marked in Figure 4.

The lower edge of the noise suppression band (*f*_L_) can be derived by finding the condition satisfying “A_u_ = cos(*k_x_W*_u_) = −1”. Among various cases, “*k_x_W*_u_ = π” is selected in this study. Because β_PDN_*W*_u_ is located far from the Brillouin zone boundary at the lower cut-off frequency, the small-angle approximation “$\sin\left(\beta_{\mathrm{PDN}}W_{u}\right)\;\approx \beta$_PDN_*W*_u_” and “$\cos\left(\beta_{\mathrm{PDN}}W_{u}\right)\;\approx 1$” can be used. Under these conditions, (6) can be rewritten as follows:(7)$$-1=1-\frac{\omega Z_{0,\mathrm{PDN}}C}{2\left(1-\omega^{2}LC\right)}\beta_{\mathrm{PDN}}W_{u}.$$

Because the wave impedance of free space (η_0_) can be expressed as $\mu_{0}c$, the lower edge of the noise suppression band (*f*_L_) can be derived from (7) and is shown in the following equation:(8)$$f_{L}=\frac{1}{\pi \sqrt{C\;\left(\mu_{0}t_{\mathrm{PDN}}+4L\right)}\;}$$

By setting different boundary condition, it is possible to estimate the upper edge of the noise suppression band (*f*_U_). It can be obtained by adopting conditions that satisfy “A_u_= cos(*k_x_ W*_u_) = +1”. Under this condition, (6) becomes the following:(9)$$\tan\left(\frac{\beta_{\mathrm{PDN}}W_{u}}{2}\right)=\frac{-\omega \;C\;Z_{0,\mathrm{PDN}}}{2\left(1-\omega^{2}LC\right)}$$

If *f*_U_ and the resonant frequency ($f_{R}$= $\frac{1}{2\pi \sqrt{LC}}$) are far away from each other, (9) can be approximated as follows:(10)$$\tan\left(\frac{\pi \sqrt{\varepsilon_{r,\mathrm{mix}}}W_{u}}{c}f_{U}\right)\approx \;\pi cf_{R}{}^{2}Z_{0,\mathrm{PDN}}\frac{1}{f_{U}}=\frac{Z_{0,\mathrm{PDN}}}{4\pi L}\frac{1}{f_{U}}$$

From (10), it is difficult to directly estimate *f*_U_. A graphical (numerical) approach is applied to estimate *f*_U_, and an example is shown in Figure 5. In Figure 5, the left-hand side (LHS) and right-hand side (RHS) of (10) are plotted in the frequency domain. The frequency where the two graphs intersect is the upper edge of the noise suppression band (*f*_U_).

The following section outlines the design and fabrication of test vehicles with thin and low-loss glass substrate and low-loss polymer as a build-up material. The test vehicles were measured to validate the proposed dispersion analysis and effectiveness of the proposed structures on noise suppression.

## 3. Verification of the Proposed Dispersion Analysis and Noise Suppression Structures

### 3.1. Design and Fabricated Test Vehicles

To validate the proposed dispersion analysis and noise suppression structures, two structures were designed and fabricated. In Figure 6, cross-sections/top views of the structures in the glass interposer test vehicles are shown. As shown in Figure 6a, this structure (Type A) had double patches to increase capacitance. Four metal layers (MLs) were needed to form this structure. In Figure 6b, a simpler structure (Type B) is shown, which only had one patch and three MLs were required. Because the copper used to form metal layers (MLs) does not adhere to the glass substrate directly, low-loss polymer was used between the substrate and MLs. In glass packages and interposers, using a low-loss polymer provides various advantages, such as additional mechanical strength, prevention of substrate cracking, prevention of moisture contact, and CTE control [35]. In each test vehicle, 25 (5 in x-direction and 5 in y-direction) unit structures, shown in Figure 6, were embedded in the PDN. The top view of the test vehicle with 25 Type A unit structures is shown in Figure 6c. Similarly, the top view of the test vehicle with 25 Type B unit structures is shown in Figure 6d. In Table 2, the physical dimensions and material properties of the test vehicles are summarized. More detailed process design rules and explanations of the structures have been described in previous works [36,37].

Each structure has different lumped capacitance (*C*) and inductance (*L*). First, the structure of Type A was analyzed to derive lumped *C* and *L*. The lumped capacitance (*C*_A_) can be derived by adding capacitance between patches and planes (*C*_pa_) and capacitance between the power/ground through-glass via (TGV) pair (*C*_TGV_) [38]. They can be summarized as follows:(11)$$C_{A}=C_{\mathrm{pa}}+C_{\mathrm{TGV}}$$
(12)$$C_{\mathrm{pa}}=\frac{\varepsilon_{0}\varepsilon_{r,\mathrm{bu}}\left(2{\left(W_{\mathrm{pa}}\right)}^{2}-\pi {\left(d_{\mathrm{TGV}\_T}/2\right)}^{2}-\pi {\left(d_{\mathrm{TGV}\_B}/2\right)}^{2}\right)}{t_{\mathrm{bu}2}-t_{m}}$$
(13)$$C_{\mathrm{TGV}}=\int_{z=0}^{t_{\mathrm{bu}1}}\frac{\pi \varepsilon_{r,\mathrm{bu}}}{{\cos h}^{-1}\left(p_{\mathrm{TGV}}/2r\left(z\right)\right)}dz+\int_{z=t_{\mathrm{bu}1}}^{t_{\mathrm{bu}1}+t_{\mathrm{sub}}}\frac{\pi \varepsilon_{r,\mathrm{sub}}}{{\cos h}^{-1}\left(p_{\mathrm{TGV}}/2r\left(z\right)\right)}dz+\int_{z=t_{\mathrm{bu}1}+t_{\mathrm{sub}}}^{2t_{\mathrm{bu}1}+t_{\mathrm{sub}}}\frac{\pi \varepsilon_{r,\mathrm{bu}}}{{\cos h}^{-1}\left(p_{\mathrm{TGV}}/2r\left(z\right)\right)}dz$$
where
(14)$$r\left(z\right)=\frac{d_{\mathrm{TGV}\_B}}{2}+\frac{d_{\mathrm{TGV}\_A}-d_{\mathrm{TGV}\_B}}{2\left(2t_{\mathrm{ub}1}+t_{\mathrm{sub}}\right)}z$$

The lumped inductance (*L*_A_) can be modeled as follows:(15)$$L_{A}=L_{\mathrm{TGV}}+2L_{\mu \mathrm{via}}$$
(16)$$L_{\mathrm{TGV}}=\int_{z=0}^{2t_{\mathrm{bu}1}+t_{\mathrm{sub}}}\frac{\mu_{0}}{\pi }{\cos h}^{-1}\left(p_{\mathrm{TGV}}/2r\left(z\right)\right)\;dz$$
(17)$$L_{\mu \mathrm{via}}=\frac{\mu_{0}\left(t_{\mathrm{bu}2}-t_{m}\right)\;}{4\pi }\left[\ln\left(\frac{4\;W_{u}{}^{2}}{\pi d_{\mu \mathrm{via}}{}^{2}}\right)+\frac{\pi d_{\mu \mathrm{via}}{}^{2}}{4\;W_{u}{}^{2}}-1\right]$$

Equation (16) has a close relationship with (13), which dominates $L_{A}$. Derivation of (16) is shown in [38]. Because microvia is not paired, the derivation of (17) is a bit different from (16). $L_{\mu \mathrm{via}}$ can be derived from the magnetic energy in the unit structure, which is known as *U*_m_ = $\frac{1}{2}$ ∫B·H *dv*. By adopting boundary condition *U*_m_ =$\;\frac{1}{2}$*I^2^L*, it is possible to derive (17) [24,39]. This relationship can be used to derive the inductance of the single TGV in the Type B structure. Because microvia is relatively shorter than the TGV pair, *d*μ__T_ and *d*μ__B_ was averaged to derive $d_{\mu \mathrm{via}}$ in (17).

The Type B structure shown in Figure 6b is simple compared to Figure 6a. The lumped capacitance (*C*_B_) can be expressed as follows:(18)$$C_{B}=\frac{\varepsilon_{0}\varepsilon_{r,\mathrm{bu}}\left(2{\left(W_{\mathrm{pa}}\right)}^{2}-\pi {\left(d_{\mathrm{TGV}\_T}/2\right)}^{2}\right)}{t_{\mathrm{bu}2}-t_{m}}$$

The inductance of the single TGV can be obtained by modifying (17). It can be summarized as follows:(19)$$L_{B}=L_{\sin\mathrm{gle}-\mathrm{TGV}}=\int_{z=0}^{t_{\mathrm{bu}1}+t_{\mathrm{sub}}}\left[\ln\left(\frac{\;W_{u}{}^{2}}{dS\left(z\right)}\right)+\frac{dS\left(z\right)}{\;W_{u}{}^{2}}-1\right]\;dz$$
where
(20)$$dS\left(z\right)=\pi {\left(\frac{d_{\mathrm{TGV}\_B}}{2}+\frac{d_{\mathrm{TGV}\_A}-d_{\mathrm{TGV}\_B}}{2\left(2t_{\mathrm{ub}1}+t_{\mathrm{sub}}\right)}z\right)}^{2}.$$

Basically, the derivation process of (17) and (19) is identical because they are both single via confined in the unit cell. Therefore, the two equations are similar. In (19) and (20), the tapered structure of the TGV is reflected. If the length of the TGV is very short and it is not tapered, the integral calculation in (19) can be simplified as in (17). The derived parameters can be inserted into the proposed dispersion analysis explained in the previous section to analyze the noise suppression band.

In Figure 7, fabricated glass interposer test vehicles are shown. Metal patches generating *C*, through-glass via (TGV), and planes are shown. Optical microscope and scanning electron microscope (SEM) were used to take images of various structures inside the glass interposer test vehicles. As can be seen from Figure 7, measurements were conducted on the probe station. Various measurements were conducted in both frequency and time domains. After verifying the effectiveness of the dispersion analysis, more structures were designed, as outlined in Section 4.

### 3.2. Verification by Measurement and EM Simulation

Measured power/ground noise couplings (*S*_21_) are plotted and compared in Figure 8. Two microprobes (Picoprobe GS type with 250 μm pitch, GGB industries Inc., Naples, FL, USA), two coaxial cables (W.L. Gore & Associates, Inc., Newark, DE, USA), and a calibration kit (CS-14, GGB industries Inc., Naples, FL, USA) were used to measure power/ground noise couplings. A vector network analyzer (VNA) (N5230A, Keysight, Santa Rosa, CA, USA) was used to measure the couplings in the PDN and validate the noise suppression band in the frequency domain up to 20 GHz. As a reference, a PDN without noise suppression structures was also measured. The distance between the two measurement ports (port 1 and port 2 shown in Figure 6c,d) was approximately 15.5 mm (≈5 ×
*W*_u_). In such cases, it is difficult to suppress the generated noise in the PDN due to low-loss substrate and polymer build-up layer. By adopting the proposed structures (Types A and B), −40 dB noise suppression bands were generated. In these frequency bands, the power/ground noise will be significantly attenuated and isolated.

Due to the double patches and paired through vias, the Type A structure had much larger lumped capacitance (*C*) than the Type B structure. The total lumped inductance (*L*) of the Type A structure was 130.80 pH, whereas it was 89.65 pH for the Type B structure. As can be seen from (8), the Type A structure had a lower $f_{L}$ compared to Type B due to larger capacitance. With larger *C*, the noise suppression band can be expanded by lowering $f_{L}$. However, the Type A structure had larger *L* compared to the Type B structure, which also lowered $f_{H}$. To achieve wider noise suppression band, higher $f_{H}$ is desired. In the following section, the impacts of various design parameters and material properties on the noise suppression band is analyzed. A design direction to achieve wider noise suppression band is also given.

In Figure 9, the measurement results are compared with simulated results. Using the 3-D EM simulator Ansys high-frequency structure simulator (HFSS) (version 2020 R2), noise coupling (*S*_21_) of each structure was estimated and compared. The measurement and simulation results showed good agreement up to 20 GHz in the frequency domain for both structures. To verify the proposed dispersion analysis, estimated stopband edges ($f_{L}$ and $f_{U}$) were compared with the simulated and measured edges. In Table 3, edges obtained by different methods are summarized and compared. The estimated edges showed good correlation with the simulated and measured edges. The accuracy of the proposed dispersion analysis for stopband edge estimation was verified. When fabricating the glass interposer test vehicles, the diameter of the TGV showed process variations associated with substrate drilling and copper plating. If the process becomes more mature, more accurate results are expected.

Compared to 3-D EM simulation, the proposed method required less time and computational resources to estimate the stopband edges. When designing the structure, the proposed dispersion analysis could effectively estimate the stopband considering the design rules and target noise band. After preliminary analysis, the structure could be designed in the 3-D EM simulator for further analysis and validation before tape-out.

## 4. Design and Analysis of Noise Suppression Structures with Various Low-Loss Materials

In this section, design directions are discussed based on the proposed dispersion analysis. The impacts of various design parameters and material properties are considered. Some candidates are chosen, and noise stopbands are estimated. Additional measurements and 3-D EM simulations are conducted to verify the impacts of the proposed structures on power/ground noise suppression and decoupling.

### 4.1. Noise Suppression Band Formulation with Various Materials

In general, broadband noise suppression is desired to cover various applications. To achieve broadband, the lower stopband edge ($f_{L}$) should be designed toward lower frequency. At the same time, the upper stopband edge ($f_{U}$) should be formed at higher frequency. Design parameters and material have significant impacts, but not all of them can be realized and adopted. It can be limited by process design rules or have a conflict with usages of advanced packages and interposers. In Table 4, the stopband expansion method by changing design parameters and impacts is summarized.

By analyzing the proposed dispersion equations, the impacts of design parameters can be easily determined. As can be seen from (8), increasing lumped capacitance (*C*), inductance (*L*), and PDN thickness will lower $f_{L}$. Adopting build-up materials with high permittivity can increase *C.* At the same time, selecting thinner build-up materials will have the same impact. However, using larger structures can increase the overall *x–y* dimensions of the packages/interposers. The *x–y* dimensions directly affect the overall fabrication yield, so this design direction is not desired. Moreover, a design direction that increases the PDN thickness should be avoided. Adding defects in power/ground planes increases *L* [31]. In terms of power integrity, this can be a good solution. However, adding defects can cause return current discontinuity issues. If such a design is adopted, routings, fan-out, and signal integrity analysis should be carefully conducted as well.

The diameter of the through-substrate via affects both stopband edges, so it is high-lighted in Table 4. When the diameter is altered, RHS of (10) is affected more dramatically, which is inversely proportional to *L*, whereas (8) is inversely proportional to $\sqrt{L}$. Moreover, the diameter of the through substrate is heavily determined by the process design rule. It is difficult to freely modify the diameter when designing packages/interposers. It is more realistic to adopt parallel through via scheme than changing the diameter to achieve lower *L.* A design that can achieve higher characteristic impedance (*Z*_0,PDN_) shifts RSH of (10) and can formulate $f_{U}$ at higher frequency. However, compared to other parameters, changing *Z*_0,PDN_ is not easy as it will have multiple impacts. The easiest method is increasing the PDN thickness, but this direction is not desired in advanced packages/interposers.

In Figure 10, dispersion diagrams with design parameters and material properties are plotted for comparison. For fare comparison, the normalized value in radian ($\beta_{x}W_{u}$) is plotted instead of the propagation constant ($\beta_{x}$). Fundamental bandgaps are formed in the bands where $\beta_{x}W_{u}$ does not exist. High-*K* material alumina can be embedded in polymer build-up layers of glass packages and interposers to increase *C* [40]. Moreover, ultrathin and high-*K* materials, such as FaradFlex substrate, can be embedded in the build-up layers of packages [18]. These materials can also be used for miniaturization of structures instead of increasing *C*. By adopting parallel trough via arrays, the suppression band is expanded toward higher frequency.

In the following subsection, additional results obtained by measurement and 3-D EM/circuit simulations are provided. The impacts of noise suppression/isolation are graphically delivered in the time domain.

### 4.2. Impacts of Power/Ground Noise Decoupling Using the Proposed Structures

In Figure 11, measured power/ground noise coupling results are plotted and compared in the time domain. A pulse-pattern generator (PPG) (Anritsu MP-1763C, Atsugi, Japan) and a digital sampling oscilloscope (Tektronix TDS800B, Beaverton, OR, USA) were used to conduct measurements in the time domain. The Type B structure explained in Section 3 was used for the experiment. Twelve gigabits per second (GB/s) clock signal (0 to 1 V, 30 ps rise-and-fall time and all ports terminated with 50 ohm) was injected to the interposer PDN (port 1 in Figure 6d) as a noise source. Frequency band of the injected noise existed in the suppression band of the Type B structure. From the noise source to the measurement location in the PDN (port 2 in Figure 6d), five unit structures existed, and the distance was approximately 15.5 mm. Without the noise suppression structure, 142 mV peak-to-peak voltage (*V*_pp_) was measured, which corresponded to 14.2% of the input voltage. When the proposed noise suppression structure was fabricated in the PDN, 51 mV *V*_pp_ was observed. Significant noise suppression/isolation was achieved by adopting the proposed structure.

Additional 3-D EM/circuit simulations were conducted to verify the effectiveness of the proposed structure. In this study, eye diagrams of the through-substrate (glass or FaradFlex) via channel under the influence of power/ground noise coupling were simulated and compared. The through-substrate via channel (victim) was designed to penetrate the PDN, and the noise source was located far away from the victim. In this scenario, the noise induced in the PDN propagates without attenuation, couples to the victim channel, and degrades SI of the victim channel (PDN-induced crosstalk). This scenario is likely to happen as there are thousands of signal through-substrate vias escaping packages or interposers, such as SerDes. Pseudo-random binary sequence (PRBS) of 2^8^ − 1, 0 V to 1.2 V, with rise-and-fall time of 30 ps and data rate (DR) of 2 GB/s was injected to the victim channel. The total length of the victim channel was designed to be approximately 14 mm, including the through via, microvia, interposer channel, and PCB channel located under the package. At the receiving location of the victim channel where eye diagrams were monitored, a capacitive termination was applied. This simulation scenario was graphically depicted in Figure 12.

In Figure 13, eye diagrams of the victim channel are plotted and compared without and with the proposed structure. In Figure 13, the Type B structure is used as a representative. In Table 5, additional results are also summarized for other structures, and eye-opening voltage, jitter, and maximum *V*_pp_ noise at logics zero/one are compared. For all three cases, eye diagrams of the through-substrate via channel were improved by adopting the proposed structures. Low-loss substrates for packages and interposers provide various advantages. However, power/ground noise must be isolated and suppressed. In this study, noise suppression was conducted by designing various structures in the PDN. The structures were analyzed and determined by the proposed dispersion analysis. Compared to full 3-D EM simulations, the proposed dispersion analysis is fast and requires smaller computational resources. Therefore, the proposed dispersion analysis is useful at the preliminary PDN design stage.

## 5. Conclusions

In this study, dispersion analysis was proposed to efficiently design power/ground noise suppression structures for packages and interposers with low-loss substrates. The mechanism of noise suppression/isolation was thoroughly explained based on the proposed dispersion analysis. By conducting the proposed dispersion analysis, the impacts on physical design parameters and material properties on the suppression band could be easily explained. To validate the proposed dispersion analysis, noise suppression structures were designed and fabricated in the glass interposer PDN and measured. It was verified that $f_{L}$ and $f_{U}$ estimated based on the proposed analysis method showed good agreement with those acquired from experiments and simulations. Compared to EM simulation, the proposed method required smaller computational resources but showed good accuracy. Various structures were designed and analyzed based on the proposed dispersion analysis. The effectiveness of the proposed structures was further validated by additional experiments and simulations in the time domain. The proposed structures suppressed power/ground noise propagation and coupling.

Low-loss substrates for packages and interposers provide various advantages, especially for high-speed signaling. However, power/ground noise must be isolated and suppressed. To solve issues, this article proposed an efficient dispersion analysis method, fabricated the noise suppression structures, and applied the structures. The proposed structures have minimal impacts on the channel routing, fan-out, and return current path. However, the structures proposed in this article require additional metal layers. These designs increase the fabrication cost. Even though they provide promising solutions toward power/ground noise issues with minimal impacts on channel properties and designs, more cost-effective designs are desired in the near future. Because the proposed dispersion analysis can be expanded to various designs, such as defects in the plane, development of a new structure based on the proposed method without increasing the number of metal layer remains the subject of work for the near future.

## Figures and Tables

**Figure 1 micromachines-13-01433-f001:**
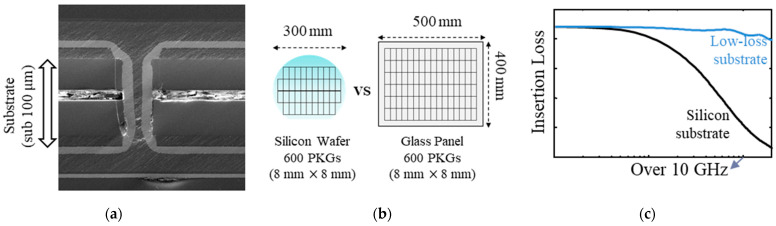
Advantages of low-loss substrate for packages and interposers. (**a**) Ultrathin (sub 100 μm) and fine-pitch metallization; (**b**) high fabrication yield, which has great potential for cost reduction; (**c**) low signal loss, enabling high-speed signaling.

**Figure 2 micromachines-13-01433-f002:**
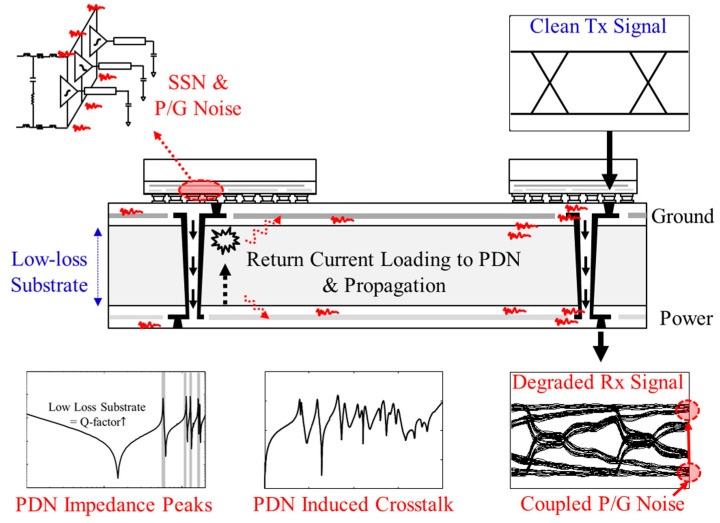
Issues related to low-loss substrate. When the noise is induced in the PDN due to SSN, P/G noise, or return current loading, low substrate loss cannot suppress it. The noise propagates along the PDN and causes SI/PI problems.

**Figure 3 micromachines-13-01433-f003:**
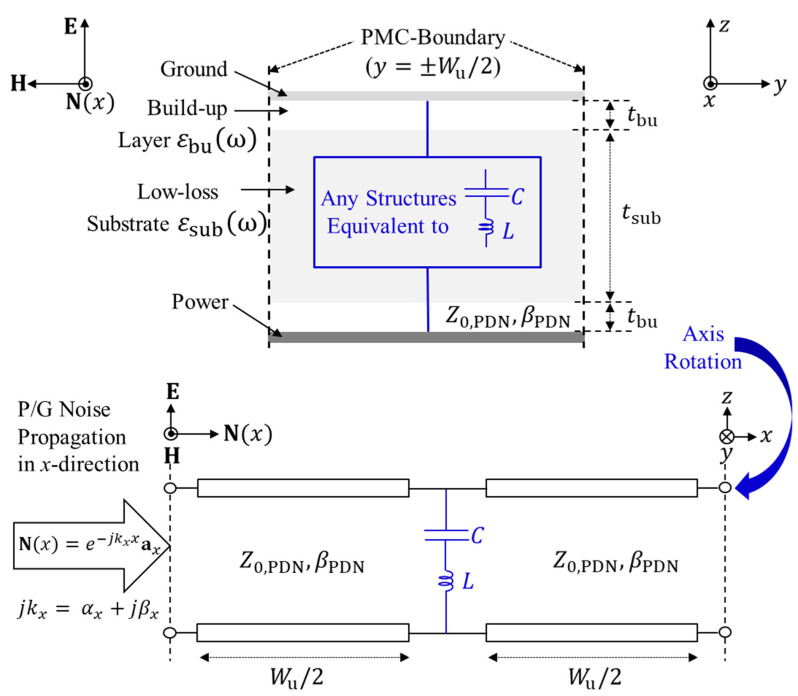
One-dimensional equivalent circuit model of the PDN with structure generating *C* and *L*. Considering the *H*-field direction, PMC boundary condition is adopted to reduce the two-dimensional PDN into one-dimensional domain in the *x*-direction. $a_{x}$ is a unit vector.

**Figure 4 micromachines-13-01433-f004:**
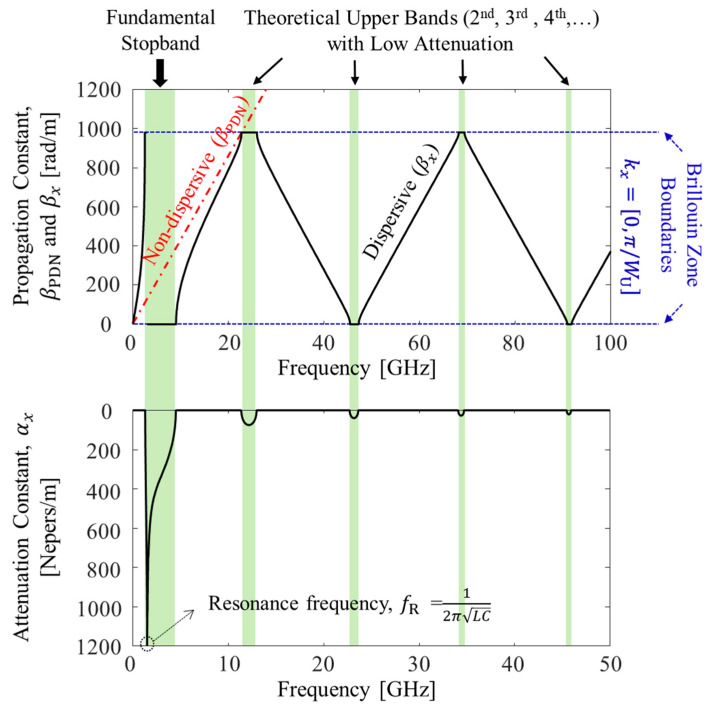
Dispersion diagrams. Without noise suppression structures in the PDN, the power/ground noise is nondispersive. With noise suppression structures, theoretical stopbands are generated.

**Figure 5 micromachines-13-01433-f005:**
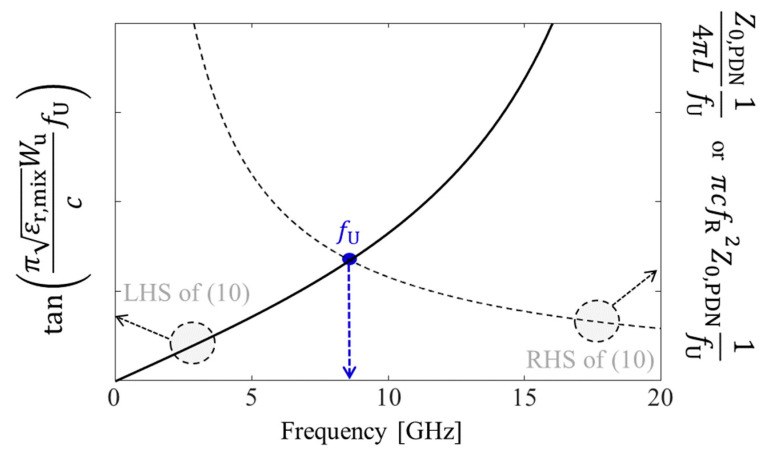
Estimation of *f*_U_ based on a graphical approach shown as an example. The LHS and RHS of (10) are plotted in the frequency domain. The intersection frequency of LHS and RHS is *f*_U_.

**Figure 6 micromachines-13-01433-f006:**
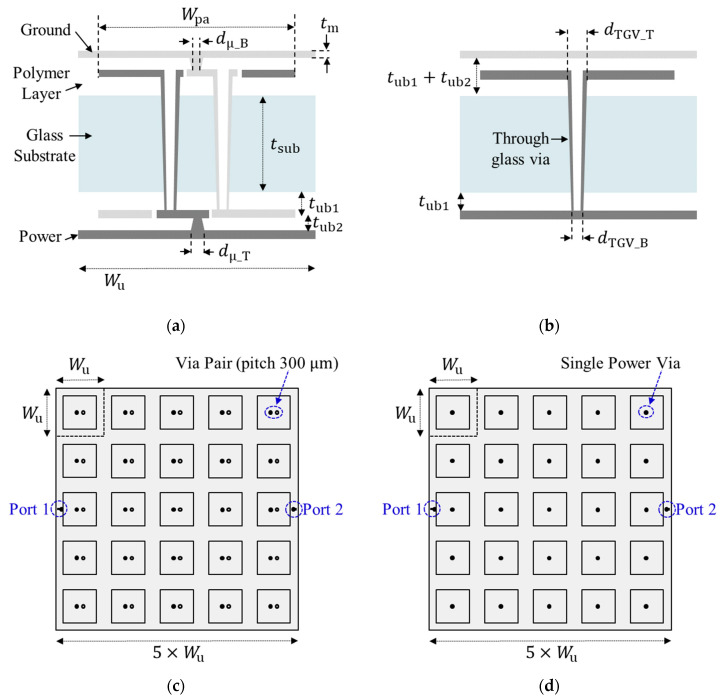
Designed noise suppression structures in the glass interposer test vehicles. (**a**) Type A: double-patch structure; (**b**) Type B: single-patch structure. Top view of the fabricated test vehicles composed with (**c**) Type A and (**d**) Type B. Each test vehicle has 25 unit structures.

**Figure 7 micromachines-13-01433-f007:**
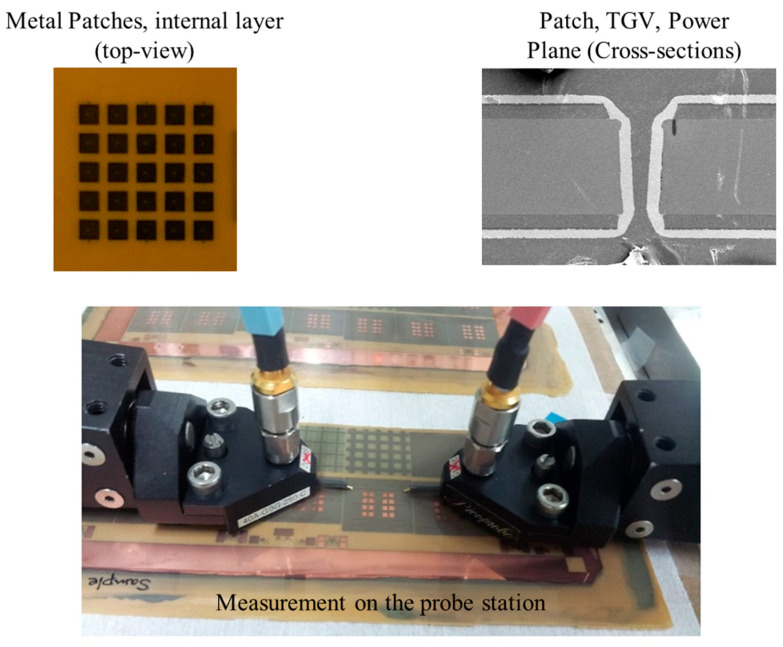
Measurement of fabricated glass interposer test vehicles.

**Figure 8 micromachines-13-01433-f008:**
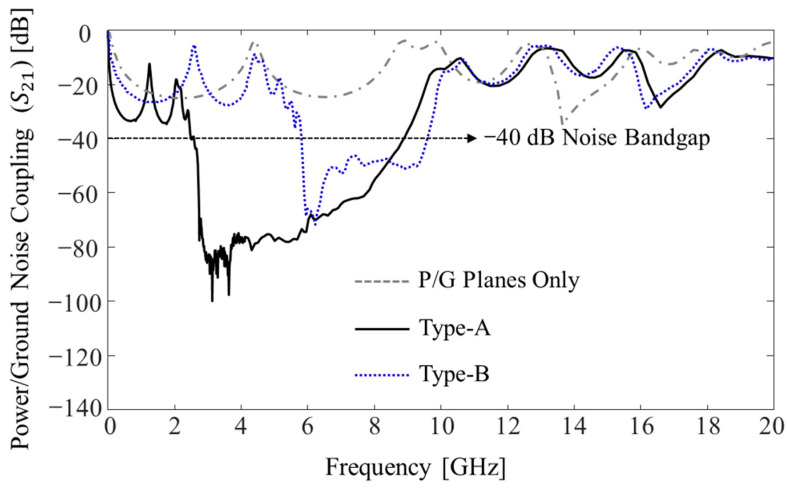
Comparison of the measured power/ground noise couplings (*S*_21_) in the PDN.

**Figure 9 micromachines-13-01433-f009:**
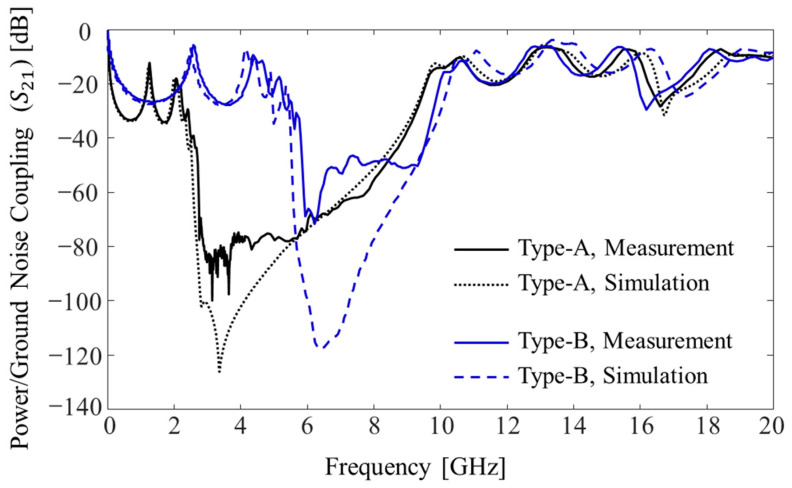
Measured power/ground noise couplings are compared with simulated results.

**Figure 10 micromachines-13-01433-f010:**
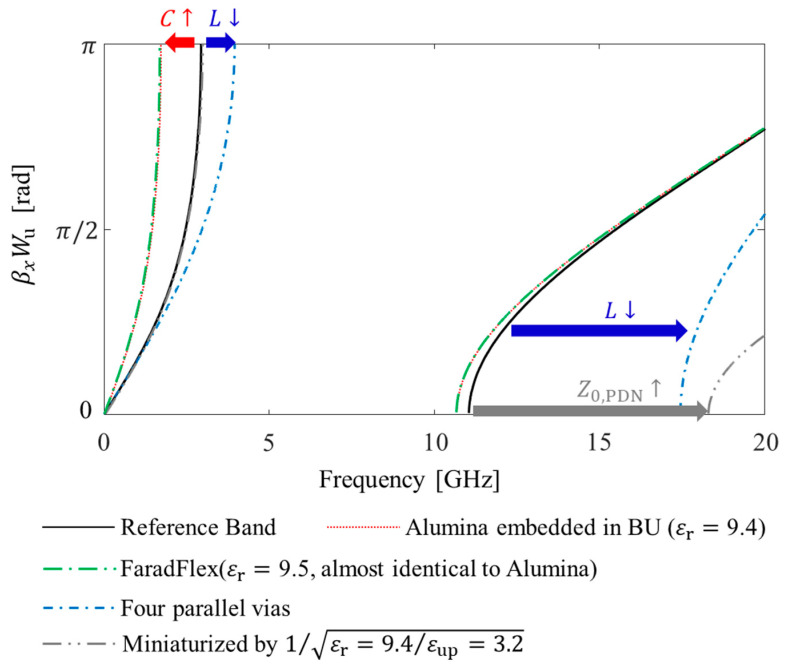
Dispersion diagrams for comparison. The impact of each parameter on stopband edge is also marked.

**Figure 11 micromachines-13-01433-f011:**
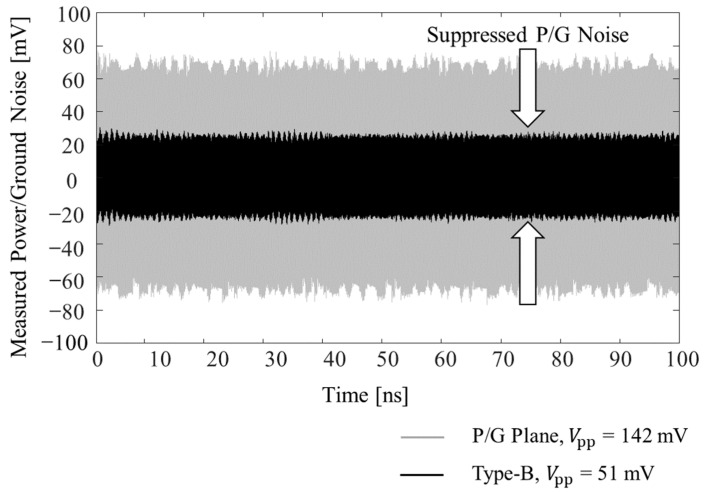
Comparison of the measured power/ground noise coupling results in the time domain without and with the noise suppression structure (Type B). By adopting the proposed structure, power/ground noise was significantly suppressed in the interposer with low-loss substrate.

**Figure 12 micromachines-13-01433-f012:**
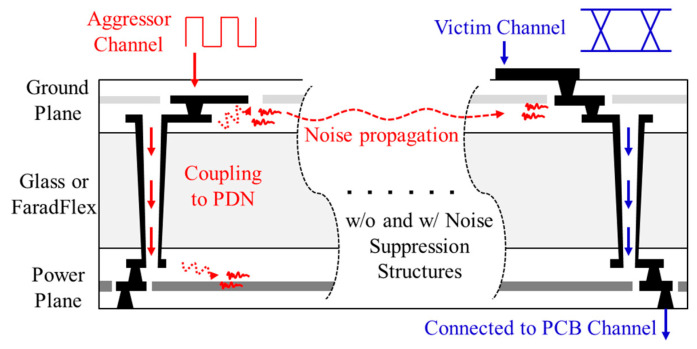
Graphical depiction of the simulation scenario. Aggressor channel escaping the package or interposer induces noise in the PDN. The noise propagates and couples to the victim channel. Eye diagrams of the victim channel are compared without and with noise suppression structures.

**Figure 13 micromachines-13-01433-f013:**
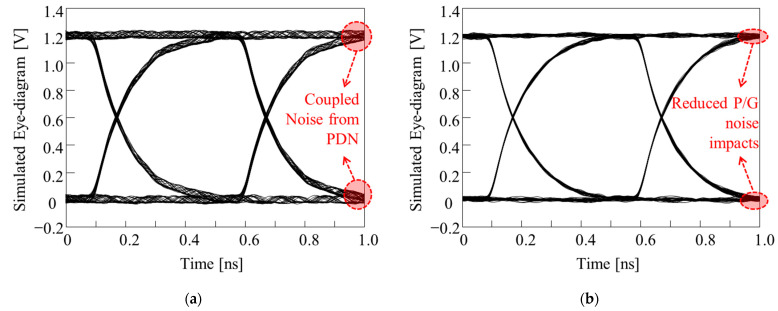
Verifications of the impacts of the proposed structure by eye diagram simulations. By adopting the proposed structure, power/ground noise coupling from the PDN to through-substrate via channel was suppressed. (**a**) P/G planes only. (**b**) embedded Type B structure.

**Table 1 micromachines-13-01433-t001:** Summary of important design parameters and values used to derive dispersion diagrams shown in Figure 4.

$\varepsilon_{r,mix}$	$Z_{0,PDN}$	$W_{u}$	*L*	*C*
4.25	9.14 Ω	3.2 mm	125.49 pH	22.831 pF

**Table 2 micromachines-13-01433-t002:** Physical dimensions and material properties of the glass interposer test vehicles.

	Symbol	Type A	Type B
Physical Dimensions	*t* _sub_	100 μm	
	*t* _bu1_	15 μm	17.5 μm
	*t* _bu2_	15 μm	35 μm
	*t* _m_	3~5 μm	4~10 μm
	*d* _TGV_T_	60 μm	100 μm
	*d* _TGV_B_	40 μm	60 μm
	*d* μ __T_	35 μm	45 μm
	*d* μ __B_	30 μm	45 μm
	*W* _u_	3.2 mm	
	*W* _pa_	2.2 mm	
	$p_{\mathrm{TGV}}$	300 μm	NA
	$\varepsilon_{r,\mathrm{sub}}$	5.3 @ 2.4 GHz	5.3 @ 2.4 GHz
	$\varepsilon_{r,\mathrm{bu}}$	3.2 @ 5.8 GHz	3 @ 10 GHz
Material Properties	*tan* δ _sub_	0.004 @ 2.4 GHz	0.004 @ 2.4 GHz
	*tan* δ _bu_	0.0042 @ 5.8 GHz	0.005 @ 10 GHz
	σ _m_	5.8 $\times \;10^{7}$σ/m	

**Table 3 micromachines-13-01433-t003:** Summary and comparison of measured, simulated, and estimated stopband edges.

Structures	Edges	Measurement	Simulation (Error)	Estimation (Error)
Type A	$f_{L}$	2.51 GHz	2.49 GHz (1.00%)	2.51 GHz (0.04%)
	$f_{U}$	8.91 GHz	8.75 GHz (1.83%)	8.59 GHz (3.61%)
Type B	$f_{L}$	5.82 GHz	5.50 GHz (5.49%)	5.87 GHz (0.86%)
	$f_{U}$	9.66 GHz	9.75 GHz (0.93%)	10.4 GHz (7.66%)

**Table 4 micromachines-13-01433-t004:** Stopband expansion method by changing design parameters and impacts.

Band Expansion	Design Parameters	Impacts	Note
$f_{L}↓$	High-*K* materials in BU	*C* ↑	
	Thin BU materials	*C* ↑	
	Larger pactes (*W*_pa_)	*C* ↑	Limited
	Increase package/interposer and PDN thickness (*t*_PDN_↑)		Not desired
	Add defects in P/G planes	*L* ↑	Limited
	Through via diameter ↓	*L* ↑	Limited & 1/$\sqrt{L}$
$f_{U}↑$	Through via diameter ↑	*L* ↓	1/*L*
	Via arrays or parallel vias	*L* ↓	1/*L*
	*Z*_0,PDN_↑(*t* _PDN_↑)	RSH of (10) ↑	Not desired

**Table 5 micromachines-13-01433-t005:** Summary of the eye diagram improvement by adopting the proposed structures.

Structures	Eye-Opening Voltage	Jitter (% of UI)	P/G Noise at 0 or 1
Type A	784 mV →838 mV	36.5 ps (7.1) →22.2 ps (4.4)	60 mV →19 mV
Type B (Figure 13)	1.08 V →1.15 V	18 ps (3.6) →7 ps (1.4)	62 mV →36 mV
FaradFlex based (Figure 10)	1.05 V →1.11 V	21 ps (4.2) →13 ps (2.6)	70 mV →39 mV

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
