# Peer review of "Design of Power/Ground Noise Suppression Structures Based on a Dispersion Analysis for Packages and Interposers with Low-Loss Substrates"

_micromachines, 2022, doi:10.3390/mi13091433_

Round 1

Reviewer 1 Report

(1) In introduction section, the nearest related works should be commented to formate a motivation for your proposal.  Or what is your research motivation?

(2) Some equations format has problems. The equations should be aligned center rather than left-aligned text. The scalar, vector and matrix is not differentiated in this paper. It is not normative.

(3)Some math symbols are not defined, such as in equation (4), “x” is ?

(4) The proposed scheme should be compared with the existing noise suppression schemes. There is no comparative results shown. How can we acquire the proof of your proposed scheme?

(5) What is the shortage of this proposed method? The future research plans should be introduced in the conclusion section.

(6) Why can your proposed scheme suppress the noise better? Can you give some theoretical analysis in the paper?

(7) What kinds of noise can be suppressed? The noise model can be talked about. There is no illustration.

Author Response

Please see the attached reply letter for reviewer 1. 

Reviewer 2 Report

The article is devoted to the important problem of noise suppression propagating along power circuits in relation to promising solutions in microelectronics. The authors proposed and implemented an effective concept of noise suppression, developed a mathematical apparatus that allows performing analytical calculations of the developed filters.

In general, the material in the article is presented clearly, consistently and clearly structured. However, some important issues for understanding are not disclosed in sufficient detail.

1) In Section 2, the author should describe in more detail the two-dimensional periodic structure under study and give a more detailed and mathematically rigorous justification for the one-dimensional approximation. It should also be indicated for which parameters the calculations performed are presented in Figure 4.

2) The author should also describe the experimental setup in more detail, in particular, indicate not only the distance between the connection points, but also the orientation of the axis connecting the connection points with respect to the translational symmetry axes of the two-dimensional periodic structure.

The cited literature is up-to-date and relevant, the author does not resort to excessive self-citation. The article is of scientific importance, the setting of the experiment is adequate to the task. The results obtained can be reproduced. The figures are understandable and properly reflect the results of calculations or measurements. The tables contain all the necessary information about the parameters of the tested structures and the properties of the materials. The conclusions made in the article correspond to the presented argumentation.

The article is of theoretical and practical interest and certainly deserves publication after improvement.

Author Response

Please see the attached reply letter for reviewer 2. 

Reviewer 3 Report

In this paper, an author presents the design and analysis of electromagnetic bandgap structures using TGV technology for power/ground noise suppression. The rigorous analysis on the stopband prediction of the EBG structures is performed. In addition, a comprehensive examination of the design parameter effects is conducted. The proposed idea is validated through measurements. The idea is interesting and the solid work is presented. However, the double stacked EBG structure and the TL-based dispersion analysis are not newly introduced. I think the main contribution of this study is the experimental characterization of the TGV-EBG structures and the experimental verification of the dispersion analysis. To emphasize the novelty, I suggest the following comments.

[1] It would be better if the author included the word of EBG structure in the main title of this paper. In this paper, the main contribution is the analysis of the EBG structure using a TGV.

[2] No references for Equation 12b, 12c, 14a and 14b are not addressed. If the extraction of these equations is the own work of the author, please explain how the equations are derived in appendix.

[3] In Figure 8, the f_L of the DS-EBG structure is substantially lowered compared to the single-patch EBG structure, while they have the similar f_H values. It would be better if the author thoroughly explained the reason why these results were obtained .

Author Response

Please see the attached reply letter for reviewer 3. 

Round 2

Reviewer 1 Report

The authors have completed the paper revision according to the reviewers' comments.

Reviewer 2 Report

The changes certainly improved the article. The article should be accepted in present form.